# Towards Precision Ophthalmology: The Role of 3D Printing and Bioprinting in Oculoplastic Surgery, Retinal, Corneal, and Glaucoma Treatment

**DOI:** 10.3390/biomimetics9030145

**Published:** 2024-02-27

**Authors:** Kevin Y. Wu, Adrian Tabari, Éric Mazerolle, Simon D. Tran

**Affiliations:** 1Division of Ophthalmology, Department of Surgery, University of Sherbrooke, Sherbrooke, QC J1G 2E8, Canada; yang.wu@usherbrooke.ca (K.Y.W.);; 2Southern Medical Program, Faculty of Medicine, University of British Columbia, Kelowna, BC V1V 1V7, Canada; 3Faculty of Dental Medicine and Oral Health Sciences, McGill University, Montreal, QC H3A 1G1, Canada

**Keywords:** biomimetics, tissue engineering, 3D printing, bioprinting, ophthalmology, oculoplastic surgery, retinal tissue engineering, corneal transplantation, glaucoma, surgical simulation, clinical translation

## Abstract

In the forefront of ophthalmic innovation, biomimetic 3D printing and bioprinting technologies are redefining patient-specific therapeutic strategies. This critical review systematically evaluates their application spectrum, spanning oculoplastic reconstruction, retinal tissue engineering, corneal transplantation, and targeted glaucoma treatments. It highlights the intricacies of these technologies, including the fundamental principles, advanced materials, and bioinks that facilitate the replication of ocular tissue architecture. The synthesis of primary studies from 2014 to 2023 provides a rigorous analysis of their evolution and current clinical implications. This review is unique in its holistic approach, juxtaposing the scientific underpinnings with clinical realities, thereby delineating the advantages over conventional modalities, and identifying translational barriers. It elucidates persistent knowledge deficits and outlines future research directions. It ultimately accentuates the imperative for multidisciplinary collaboration to enhance the clinical integration of these biotechnologies, culminating in a paradigm shift towards individualized ophthalmic care.

## 1. Introduction

At the vanguard of ophthalmological innovation, biomimetics via 3D printing and bioprinting is paving the way for transformative advances in the field. These biomimetic strategies not only replicate the structural intricacies of ocular tissues but are also unlocking groundbreaking possibilities in oculoplastic and orbital surgery, retinal tissue engineering, corneal transplantation, and glaucoma intervention. This review provides a comprehensive examination of the current and potential applications of these technologies in ophthalmological practices and research.

In the realm of prototyping and simulation, we discuss the development of training models for surgical practice, highlighting the state-of-the-art models currently in use. The importance of preoperative surgical planning and simulation tools is underscored. The review also delves into specific applications within ophthalmology, including oculoplastic and orbital surgery. Here, innovations such as ocular prostheses, orbital implants, nasolacrimal stents, adjustable eyelid crutches, drug-loaded punctal plugs, and their clinical significance are explored.

The review also covers the advancements in retinal applications through the use of 3D printing and bioprinting, particularly in the development of macular buckles and retinal tissue engineering. Similarly, in corneal applications, the focus is on the fabrication of specialized contact lenses and corneal tissue engineering. These advancements offer potential improvements over traditional treatment methods in terms of customization. The article further discusses the integration of these technologies in glaucoma treatment, highlighting the development of minimally invasive glaucoma surgery (MIGS) devices and drug-eluting implants.

Our literature review encompasses primary studies from 2014 to 2023, providing a thorough analysis of the evolution and current state of these technologies in ophthalmology. Unlike previous reviews which focused predominantly on benchwork data, this study distinguishes itself by its comprehensive approach. It evaluates not only the fundamental science behind biotechnologies but also their clinical implications, scrutinizing conventional treatments and methods alongside their challenges and drawbacks. This review articulates the advantages of 3D printing and bioprinting in addressing these challenges, compares these technologies to conventional methods, and identifies limitations and barriers to clinical translation. Furthermore, it elucidates knowledge gaps and proposes directions for future research, grounded in both basic science and clinical insights.

The review gives a nuanced emphasis on the transformative potential of 3D printing and bioprinting in ophthalmology, acknowledging the challenges ahead, and highlighting the need for continued interdisciplinary collaboration to advance these technologies from the laboratory to clinical practice. This dual approach fosters a deeper understanding of the trajectory from preclinical studies to clinical applications, facilitating a more effective translation of these innovative biotechnologies into everyday clinical use.

## 2. Applications of 3D Printing in Oculoplastic and Orbital Surgery 

### 2.1. Applications for Orbital Implants and Prosthesis

Orbital implants and prosthetic devices are essential in re-establishing facial balance and aesthetic appeal following surgical interventions such as evisceration and enucleation. The evisceration technique entails removing the internal contents of the eye but preserving the sclera, extraocular muscles, and optic nerve, thereby keeping the structural relationships of the muscles, globe, eyelids, and fornices [1]. This preservation facilitates improved motility of the prosthesis and simplifies its fitting process [2,3,4]. Evisceration is contraindicated in cases where intraocular malignancy is suspected due to its conservative nature. Conversely, enucleation involves the excision of the entire eye, sparing the surrounding orbital tissues. It is predominantly employed for treating intraocular cancers such as retinoblastoma and choroidal melanoma, particularly when they do not respond to alternative treatments [1]. This method allows for a thorough histological examination, minimizing the risk of tumor cell spread as well as sympathetic ophthalmia (although this is debatable).

Orbital implants serve a dual purpose: they compensate for the loss of orbital volume and preserve the orbit’s architecture [5]. Additionally, these implants facilitate movement in the ocular prosthesis placed above them, contributing to the restoration of eye movements and promoting a more natural look [5]. Customization of the prosthesis occurs approximately four weeks after surgery, tailored precisely to fit the dimensions of the patient’s conjunctival fornices and overall anatomy [6]. This customization is critical for optimizing cosmetic outcomes, as it ensures the prosthesis closely resembles the contralateral eye in terms of color, size, and positioning, thereby enhancing the overall aesthetic effect.

Currently, providing patients with personalized prostheses is a tedious procedure composed of many precise steps [7]. Three-dimensional printing shows great potential in minimizing imperfections and enhancing fabrication rates. Using computer-aided design (CAD/CAM) and rapid manufacturing (RM) technology, 3D-printed prostheses are made possible. 

Ruiters et al. (2016) used patient-specific imaging provided by cone-beam computed tomography systems (CBCT) to reconstruct and design a personalized digital prosthesis that was then 3D-printed [8]. Compared to the conventional method of obtaining patient orbit structure through direct mold impression, imperfections relating to soft tissue impression that could have otherwise occurred in the modeling process are mitigated [9]. Additionally, this technique allows for a reduction in production time by eliminating the molding step. However, radiation exposure inhibits certain populations, such as children, from accessing this service [8]. 

Alam et al. (2017) provide a possible solution to radiation exposure by maintaining the use of a direct impression of the orbit, digitizing the mold, and fixing imperfections virtually before printing the prosthesis. Taking it a step further, the anterior of the virtual model is then designed to be hollow, minimizing the weight of the final product. Patients reported better comfort levels when using this prosthesis as compared to their conventionally made prosthesis. Fabrication time and weight are also greatly optimized [10]. 

Studies have also explored the potential of complete automated ocular prosthesis manufacturing, using technology to print the prosthesis itself and the aesthetic design of iris color and blood vessels [11,12,13,14]. Using a photograph of the contralateral unaffected eye, the physical features of the eye are then placed on the 3D model by sublimation. This method allows for a significant decrease in production time as pattern design would otherwise be hand-drawn by an ocularist. Moreover, the blueprint of patient-specific prosthesis can be safely stored, allowing for quick replacement if needed [11,12,13]. This process is simple and advantageous for beginner ocularists, as conventional ocular prosthesis design turnout depends on an ocularist’s experience [15]. 

Further studies have explored the use of 3D printing for both the printing of the prosthesis and its pattern design [13,14]. The same benefits apply, though the color scheme is heavily limited by the color selection that the 3D printer offers [13]. Moreover, as ocularist experience in automated pattern painting is less significant, experience in 3D graphics and modeling becomes essential in producing these prostheses [14]. An important factor in the 3D fabrication of such products is the use of biocompatible material, suitable for long-term wear by the patients. 

Kim et al. (2021) studied the biocompatibility of their 3D resin-printed, polymethyl methacrylate-coated prosthesis, which showed biological and physicochemical safety [11]. However, there remains the need to establish a gold-standard material when producing 3D-printed prostheses. Beyond ocular prosthesis, 3D-printed ocular resin implants have also been explored and showed great results 12 months post-op without any systemic effects observed [16]. Three-dimensional printing models can also act as guides for surgical procedures. Indeed, Weisson et al. (2020) used the facial topography of patients to digitally construct an orbital prosthesis that is then 3D-printed. This product acts as a mold from which silicone is cast to produce the final prosthesis. The results showed great potential and required little to no consultation from prosthetists or ocularists. However, printing and silicone casting remain rate-limiting steps of this procedure [15]. 

Similarly, in the context of osseointegrated implants, virtually designed guides with holes indicating the location of drilling of implants are 3D-printed and overlaid on the surgical site, allowing for reduced surgery time and increased surgical precision, resulting in a good outcome [17]. Furthermore, 3D printing can be advantageous to mitigate surgical complications. For example, orbital migration can reduce aesthetic outcomes and cause surrounding tissue fibrosis. In this context, Dave et al. (2018) reported a case study where a 3D-printed implant was incorporated into the migrated implant eye to recenter the orbit. Though the procedure left a bulge which the authors deemed normal for this type of intervention, the application was successful [18]. 

Lastly, 3D printing in the context of conformers for fornix deepening post-enucleation is possible [19,20]. Currently, the standard of care offers non-personalized conformers of various sizes. However, well-fitted conformers are essential to preserve the shallowing of the fornices; hence the advantage of 3D-printed conformers, which allow for personalized conformers and custom fixation holes [19]. 

As various means of ocular prosthesis were explored, the advantages of this method seem to include the reduced production time [8,10,12,13,15,17], personalization [12,13,17,19,20], and access [11,15,16]. By adapting to patients’ personal anatomical traits, this can lead to overall positive outcomes in terms of aesthetics and function. However, there remain gaps to be addressed. In fact, most studies discussed were based on case reports of a few patients [8,12,15,17,18,19] or pilot studies [10,11,21]; hence the importance of large-scale and long-follow-up studies. Moreover, the method of topography, producing a detailed virtual replica of the patient’s anatomy, remains complicated. The use of CT or CBCT exposes patients to radiation, which can be contraindicated in some, whereas MRI scans can cause further artifacts [8]. Alternatives such as hand-held blue-light-based 3D scanners [15] and light-intensity 3D scanners [12] have been explored and showed positive final results. 

Finally, the production material of the 3D-printed models is an area needing further research. As specific materials are starting to be deemed biocompatible [11], it is important to explore all possible alternatives while considering the effect of such material on patient comfort and overall experience [21].

Table 1 provides an overview of the application of 3D printing in the creation of ocular prostheses, detailing the direct use of this technology in making various prosthetic devices.

### 2.2. Use of Orbital Implants in the Repair of Orbital Floor Injuries

Orbital trauma, frequently encountered in emergency consultations due to trauma, is commonly associated with injuries to the facial bones and soft tissues, culminating in orbital floor fractures, also known as blowout fractures [22]. These fractures primarily occur when the orbit is struck by objects exceeding its opening in size. The underlying mechanisms, as described by the hydraulic theory and the buckling theory, involve a sudden rise in intraorbital pressure or deformation of the orbital rim, respectively. Such dynamics lead to fractures that may result in the displacement and entrapment of orbital contents into the maxillary sinus [23,24,25,26]. 

The management of orbital floor fractures often involves a period of observation to allow an edema and orbital hemorrhage to resolve naturally. However, surgical intervention becomes necessary in certain cases. Surgical indications typically encompass persistent diplopia, particularly with restricted upgaze and/or downgaze movements within 30 degrees of the primary gaze, along with positive results on forced duction testing [22,27,28]. These findings indicate a likely functional entrapment affecting the inferior rectus muscle. Additional surgical indications include enophthalmos greater than 2 mm, which is deemed cosmetically unsatisfactory, and fractures that compromise at least half of the orbital floor’s integrity [22,27,28].

The favored surgical technique for addressing orbital floor fractures typically utilizes an inferior transconjunctival incision. This approach includes lifting the periorbita away from the orbital floor, freeing any herniated extraocular muscles and fat tissue entrapped by the fracture, and securing an implant to cover the fracture site, thereby averting further herniation [29,30,31]. The primary goal of reconstructing the orbital wall is to re-establish the normal anatomical configuration of the internal orbit.

Orbital implants play an important role in post-traumatic orbital reconstruction, ensuring anatomical and orbital function and aesthetics. Current materials used in such implants include autologous, allogenic, and alloplastic materials such as bone, lyophilized dura, and titanium respectively. These require surgical experience to bend and size the implant intraoperatively with great precision, as larger-than-needed dimensions could cause globe displacement or extraocular muscle dysfunction. 

Current investigations into 3D printing technology explore its viability for crafting personalized orbital implants, with a significant body of research focusing on utilizing 3D models for implant customization [32,33,34,35,36,37,38,39,40,41,42,43,44] and for the creation of directly printed implants [45,46,47,48,49,50]. Chai et al. (2021) demonstrated the efficacy of using patient-specific 3D-printed models as templates for intraoperative adjustments to implants, noting a marked decrease in both surgery duration and related complications [32]. This process involves digitally reconstructing the damaged orbit and overlaying this model onto the implant, serving as a precise guide for surgical modifications [33,34]. Moreover, handling autologous bone grafts, known for their delicacy and challenging manipulation, has been improved through the use of 3D-printed templates. Vehmeijer et al. (2016) highlighted the benefits of this approach, including increased precision, operational efficiency, and cost-effectiveness [34]. 

Research has also delved into the use of 3D-printed models as molds for orbital implants, offering a distinct approach from templates. In this method, the implant is shaped directly against the 3D model, ensuring a precise fit. This technique of pre-operative shaping has been extensively studied with materials such as titanium mesh [35,36,37,38,39] and polyethylene plates [40]. The consensus from these investigations suggests that utilizing 3D-printed molds improves surgical outcomes and is advantageous over conventional preparation techniques.

For example, Loneac et al. (2021) explored this idea in comminuted zygomaticomaxillary complex fractures, which resulted in significantly better orbital volume restoration and symmetrization when compared to traditional free-hand bending. Moreover, Sigron et al. (2020) highlighted the reduction in surgery time as well as its cost-efficiency, though this can vary amongst institutions. Personalized pre-bent implants also greatly diminish the need for intraoperative fitting, further reducing surgery time [39,40]. Beyond fractures, the mold method can be applied to orbital pathologies. Mourits et al. (2016) used a 3D-printed model of a patient’s orbital cystic growth. This allowed for direct 3D visualization of the patient’s anatomy. Further, it allows for intraoperative implant manufacturing and modifications compared to direct 3D printing where material can only be removed and not added [43]. 

Finally, a 3D-printed pressing apparatus designed to press a real implant between two 3D models of the implant is explored. Rather than the surgeon cutting or molding the implant, they hold and press the apparatus to obtain a ready-to-use personalized implant [41,42]. Though not directly compared to 3D-printed templates or molds, this technique shows excellent outcomes through quantitative [42] and qualitative [41] measures. However, these pressing apparatuses were only evaluated in small sample sizes. 

When looking at direct 3D printing of orbital implants, Kim S.Y. et al. (2019) demonstrated adequate postoperative results, mitigating infection or inflammatory response while maintaining functional outcomes [46]. Choi et al. (2022) applied direct printing as an option for aesthetic reconstruction in patients with previous facial fractures presenting with late enophthalmos and hypoglobus, demonstrating the possible benefits of 3D implants in postoperative care [47]. 

Direct printing of implants in complex fractures is also deemed successful [44,48,49,50]. To maintain the structure and function of complex fractures, a jigsaw puzzle technique is employed where small implants are 3D-printed and attached in vivo to construct an overall larger implant. This allows for personalized and uneventful postoperative results. However, results are based on case reports; thus, the need for larger studies to examine the reliability of this application. 

Kim J.H. et al. (2020) [45] moved away from personalized implants to investigate the potential of a standard inferomedial orbital strut implant to mitigate the manual bending of traditional implants and the production time and cost of personalized 3D-printed implants. Through collecting data from 100 adult cadavers, a standard fit implant showing satisfactory results in simulated and real patients was created [51]. Most experiments relating to 3D printing highlight its ability to customize implants, though this study reveals a situation in which printing standard, non-personalized implants may yield similar benefits. 

In summary, 3D printing provides several benefits in terms of accuracy and surgical outcomes (Table 2), with most orbital implant guides made of titanium [35,36,37,38,39] and direct models made of polycaprolactone [45,46,47]. Nonetheless, several gaps remain. For instance, when it comes to the best way of digitally constructing the affected eye, some opted for mirroring the unaffected orbit [35,36,37,48,49,50]. This method was criticized by Tel et al. (2019), since human facial anatomy is not completely symmetric; thus, relying on the contralateral anatomy may cause design defects [33]. Instead, they applied spline interpolation techniques. Others approached this by using reconstruction softwares to build adequate personalized implants [42], while some utilized a mixture of virtual and manual planning [39,43]. Regardless of the technique used to obtain a final implant, all showed adequate function when implemented. From templates to direct printing, there exists a wide range of 3D printing applications in this domain that have shown great potential compared to conventional techniques. As these methods are analyzed further, studies comparing these new procedures with one another can better highlight each technique’s unique benefits and weaknesses.

### 2.3. Applications for Assorted Ophthalmic Procedures (Nasolacrimal Stents, Drug Delivery, and Eyelid Crutches)

3D printing has been explored in various domains relating to oculoplasty outside of surgery. Sun et al. (2019) investigated the application of 3D printing in addressing ptosis through universal eyelid crutches as a low-cost, non-operative treatment choice. Compared to current personalized eyelid crutches, they introduced an adjustable and easily removable crutch that can be attached to various eyeglass designs. Though only a case report and there remains a need to directly compare these to traditional crutches, Sun et al. (2019) highlight the potential use of this 3D model in developing countries, where expensive personalized crutches may not be an option [52]. Furthermore, this study differentiates itself from others by underlying the fact that a standard, universal apparatus design may sometimes be more beneficial than custom-fitted ones. In dry eye disease, 3D-printed punctal plugs have been studied to treat dry eyes while simultaneously delivering drugs and increasing patient compliance compared to traditional topical drugs [53,54]. Xu et al. (2021) studied a 3D-printed drug-loaded punctal plug in vivo, which showed extended release characteristics and the absence of drug-photopolymer interaction when using dexamethasone and polyethylene glycol [54]. Similarly, in the context of nasolacrimal stent, 2-Hydroxyethyl methacrylate hydrogel resin was found to be biocompatible in in vivo animal studies [55]. As research on this topic continues to evolve, clinical studies on various drug and printing material interactions remain to be studied. Radiotherapy eye shields, which are necessary when treating orofacial tumors, are traditionally personalized through open-eye impressions with the need for anesthesia. Using a commercial surface scanner to digitize patient anatomy eliminates direct contact with the eye and allows for the fabrication of 3D-printed shields using readily available biocompatible material, though post-scan digital modifications were needed due to scanner flash causing scanning defects of the iris [51]. Despite the possibility of scanning difficulties needing post-scanning adjustment, this method is a possible alternative to radiation exposure by CT scans. Finally, 3D printing bridges the fields of ocular pathology and engineering, creating novel solutions to important pathologies. For example, Yang et al. (2022) studied the use of a drilling 3D microrobot as an alternative to nasolacrimal duct probing to remove primary acquired nasolacrimal duct obstructions. This technology has shown great potential in the early stages of phantom model testing, or objects designed to imitate the human body [56,57]. In essence, there exist great opportunities for 3D printing in various non-surgical aspects of ocular disease. However, as of now, the majority remain developmental [52,53] in vivo [54,55], or simply case studies [51,53]. As novel research on the topic of 3D printing in ocular disease continues to emerge, large-scale clinical studies to properly depict the outcomes of such new technology remain important.

Table 3 summarizes the advances in 3D printing for nasolacrimal stents. Table 4 details the use of 3D printing in non-surgical ophthalmic applications, such as drug delivery through punctal plugs, ptosis treatment with eyelid crutches, and radiation protection with eye shields. Table 5 presents a summary of case report data and clinical study data concerning 3D printed ocular and orbital prostheses, conformers, and repair of orbital floor fractures. 

## 3. Retinal Applications

The clinical uses for bioprinted and 3D-printed materials in retinal medicine are vast. These uses include retina-targeted drug delivery platforms, retinal tissue models for in vivo research, ophthalmologist training, and retinal prosthetics for treating retinal degenerative disease. The 3D printing and bioprinting of ocular devices, scaffolds, and tissue models offer benefits including product standardization and fine tailoring of the shape, porosity, and curvature of products. However, there are many obstacles left to overcome when attempting to create a model that emulates the anatomical complexity of the retina (Figure 1).

### 3.1. Study of Retinal Disease through Retinal Modeling

The retinal pigment epithelium (RPE) is a layer of single, polygonal cells sandwiched between photoreceptor cells and Bruch’s membrane at the outermost retinal layer. The RPE forms the outer blood–retinal barrier through tight junctions [59] and regulates the passage of substances in and out of the retina. RPE damage is implicated in degenerative diseases including retinitis pigmentosa, wet and dry age-related macular degeneration (AMD), and Stargardt’s disease. A well-developed in vitro retinal model can be used to study such retinal degenerative diseases, elucidate pathophysiological mechanisms for disease development, or test new pharmacotherapy. Due to the complex interplay of junction proteins and hormonal signaling between retinal layers, this is a challenging task. An ideal model needs to have cells with appropriate polarity, tight junction protein expression, proper attachment to Bruch’s membrane, and the release of hormones such as human vascular endothelial growth factor (hVEGF) at physiologically similar levels, among other things. Several bioprinting technologies exist and can be used to help develop these retinal models [60].

Figure 2 illustrates various bioprinting technologies that are employed in the development of retinal and corneal applications, detailing different methods under the categories of vat polymerization, extrusion-based, and jetting-based bioprinting.

Shi et al. developed a bilayer, photoreceptor–retinal tissue model using microvalve bioprinting. ARPE-19 and Y79 cells were either bioprinted or manually seeded onto scaffolding. Manual cell seeding forms aggregates with uneven cell layers while microvalve bioprinting develops an organized cell monolayer with high zonula occuldens-1 (ZO-1) and claudin-1 (C-1) expression. The orderly distribution of cells using bioprinting is thought to enhance the expression of these cell–cell adhesion proteins [61]. Tight junction development is needed for an effective RPE model given its role in the blood–retinal barrier. With these characteristics, this model demonstrates potential use as an in vitro model for retinal disease research. This study also clearly demonstrates a common benefit of most bioprinting technologies compared to manual seeding—the speed at which cell layers can be seeded onto scaffolding. Like ARPE-19 and Y79 cells, the growth and differentiation of Müller (retinal glial) cells depends on the structural characteristics of the growth medium. Jung et al. found that physiological differences exist between 2D cultured & 3D bioprinted Müller cells, specifically in potassium and water channel expression, and cell cytokine and growth factors [62]. Given cell scaffolds and bioprinting techniques can directly affect cell protein expression, the expression of important proteins should be compared between in vitro and in vivo retinal cells for potential differences.

Proper RPE tight junction development is also likely to be more dependent on bioprinting than the presence of scaffolding. Masaeli et al. developed a scaffold-free, dual-layered retinal cell model using inkjet-based bioprinting composed of an RPE and photoreceptor layer. Similar to the research by Shi et al., high ZO-1 and C-1 expression was noted, but hVEGF, an endothelial angiogenic and vasopermeability factor, was also detected [63]. Kim et al. furthered this idea by hypothesizing that RPE printing onto polymer scaffolding is insufficient to lead to full RPE maturation. A Bruch’s membrane-mimetic substrate (BMS) was found to have more influence on RPE layer development versus an uncoated Transwell scaffold. The layered BMS-RPE on a Transwell scaffold had higher ZO-1 and phototransduction enzyme expression and higher transepithelial electrical resistance (TEER), a measure of barrier function. Furthermore, there is potential for the use of BMS-RPE for in vivo transplantation as the cell layers can easily be removed from the underlying scaffold and transplanted subretinally and subcutaneously [64].

In a recent study, Song et al. designed a more comprehensive, multilayered retinal model with the development of an outer blood–retinal barrier (oBRB) tissue composed of endothelial cells, pericytes, and fibroblasts bioprinted on the basal side of a scaffold with an RPE monolayer on top. The oBRB tissue acts as an excellent model for retinal degeneration in vitro. Both wet and dry AMD phenotypes were induced in the oBRB tissue—complement activation manifests as dry AMD (drusen development and choriocapillaris breakdown) while HIF-α stabilization or STAT3 overactivation leads to the development of type-I wet AMD (choriocapillaris neovascularization) [65]. Three-dimensional bioprinted retinal tissue models can also be used to study external environmental factors on ocular health. Kim et al. found that oxidative stress induced by cigarette smoke can disrupt RPE tight junction strength and blood–retinal barrier function [66]. This information can be used to further explore causative links between smoking and retinal degeneration.

Orientation and polarity are also important in the survival and development of many types of ocular cells, including retinal ganglion cells (RGCs). RGCs help project retinal signals to the brain’s visual centers. Kador et al. [22] combined a technique known as electrospinning [23] with thermal inkjet 3D bioprinting to create a nanoporous scaffold with RGC growth. The scaffolding helped correctly orient a proportion of growing neurites (71.9% of axons, 49.3% of dendrites) [22], correctly demonstrating that the nanopores created by electrospinning create an acceptable scaffold for future retinal tissue growth. Without in vivo testing, we cannot determine if this model may provide clinical benefit for diseases requiring RGC implantation. 

### 3.2. Retinal Cell Delivery Scaffolds

Age-related macular degeneration (AMD) is a degenerative retinal disease that affects the retinal macular region at the RPE level. Drusen formation occurs between Bruch’s membrane and the RPE, and depending on the extent of drusen development—based on size, and number—AMD may be diagnosed and categorized as wet (exudative/neovascular) or dry (non-exudative/non-vascular) AMD [67]. The mainstay of wet AMD treatment is intravitreal anti-vascular endothelial growth factor (anti-VEGF) injection. Poor adherence to therapy is common due to the need for inconvenient and irritating repeated intravitreal injections [68]. Repeated intravitreal injections also increase the risk of endophthalmitis [69] and physical tears in the RPE layer [70]. Even with optimal adherence, visual function still deteriorates gradually over time [71] with anti-VEGF therapy, so significant room for improvement in AMD therapies still exists. A school of research is currently exploring cell transplantation as a potential treatment for degenerative retinal diseases like AMD in patients unresponsive or resistant to conventional pharmacotherapy. Furthermore, 3D-bioprinted scaffolds have the potential to replace or augment the diseased RPE.

Retinal cells cannot regenerate [24], meaning diseases of the retina are degenerative, worsening over time without improvement. Pharmacologic therapy has made significant progress with the commercialization of anti-VEGF therapies for conditions like AMD [25]; these therapies are not curative, however. New research on these diseases focuses on cell replacement therapy to replace damaged retinal cells [26], but barriers preventing their clinical use still exist, including effective ways to deliver the cells to the retinal layer. Retinal cell delivery through intravitreal injection has risks of unpredictable cell viability, uncontrolled cell proliferation in the vitreous cavity, macrophage infiltration, and ocular inflammation [27]. The use of 3D-printed biocompatible scaffolds to grow and deliver cells could be a viable option. The ideal scaffold should encourage cell growth with the correct orientation and polarity.

Due to poor printing resolution, 3D printers can act as the constraining factor when designing materials. Poor resolution prints can result in flat, ‘two-dimensional’ scaffold structures which encourage unorganized cell growth. Two-photon polymerization (2PP) 3D printing allows better resolution than most other 3D printing techniques. Using 2PP, Worthington et al. developed organized, porous scaffolds seeded with induced pluripotent stem cell (iPSC)-derived retinal progenitor cells. Maximum cell loading occurred in the 25 μm diameter pores with correct cell polarity and cell viability throughout. Seeding cells on a physical support substrate is hypothesized to be the reason for improved cell viability and proliferation compared to bolus injections of cell suspensions [28]. The challenge with 2PP is material degradation due to the high-energy lasers used in the process [29] which may not be suitable for all types of scaffolding materials. In vivo subretinal implantation of a scaffold device similar to the one developed by Worthington et al. [28] was developed by Thompson et al. [30]. The implantation in a porcine model of retinitis pigmentosa showed no sign of inflammation, toxicity, or infection after one month. The potential for 2PP to develop clinically useful 3D cell-delivery scaffolds for the treatment of retinal diseases exists, but the extent of the research is limited as the majority of research on RPE cell transplantation involves non-3D-printed scaffolding [31,72].

### 3.3. 3D-Printing Assisted Macular Buckling 

Macular buckle (MB) is a surgical technique used to reshape the posterior segment wall when treating high myopia. The axial elongation of the eye in untreated MB can cause the stretching of ocular structures and eventually retinal degeneration [73]. Zou et al. used optical coherence tomography, 3D modeling, and 3D printing to assist in the treatment of myopic foveoschisis (MF) via MB surgery. MRI imaging data was used to reconstruct a 3D model of the patient’s eye and a 3D print of the model allowed clinicians to design a titanium stent for MB surgery. The clinical endpoints for MF post-MB surgery were positive, including significant visual acuity improvement and axial length reduction at 1 year. Three-dimensional modeling also helped with MB stent positioning. It is notable, however, that MB surgery typically uses commercial stents and Zou et al. make no comparison between the ease of use or effectiveness of printed stents versus commercial stents. Furthermore, the stent modeling and printing required additional time, leading to prolonged hospitalization and additional costs to the patient [74]. The need for a custom-designed MB stent is still uncertain given the lack of comparison to existing commercial stents.

### 3.4. Retinal Drug Delivery Platforms

Three-dimensional printing technology shows promise in its ability to create biodegradable scaffolds for sustained drug release in the retina. This could be used as an alternative method for drug delivery in diseases like wet AMD which require repeat intravitreal injection of anti-VEGF agents [75], risking the development of endophthalmitis and other complications. Won et al. developed a 3D-printed rod with a drug depot containing bevacizumab and dexamethasone. Structurally composed of a polymeric shell and a hydrogel core structure each with a separate drug depot, this device can release two drugs with differing release kinetics in the vitreous humor. Significant in vivo reduction of angiogenic and inflammatory markers (isolectin B4 and CD45, respectively) were noted two weeks post-intravitreal injection of the depot device in a rat model. In vivo studies showed more long-term sustained drug release using the rod versus direct intravitreal injection. Although this technology may reduce the need for repeated intravitreal injection, there was no mention of the rod disintegrating in the vitreous humor, whether the device needs to be removed, or the ability of the drug depot to release past 60 days [76]. 

### 3.5. Ophthalmologist & Patient Training

Three-dimensional printing technology can be used to develop and improve ocular models for both ophthalmologist surgical training and patient education. Currently, few realistic ocular models for ophthalmologist retinal laser surgery training exist. The majority of current models have rigid casings and a limited view of the fundus due to poor structural designs differing from actual ocular anatomy. Pugalendhi et al. developed a 3D ocular model to correct these two differences. The resulting model improved the fundus viewing area by almost 17%, creating a more realistic model to help with surgical training [77]. In patient education, the complex mechanisms of ocular disease can be difficult for patients to visualize and challenging for physicians to explain. Yap et al. developed 3D-printed models of the retina to demonstrate the pathophysiology of AMD and how the different layers of the retina are affected by the disease [78]. Although outcomes of patient understanding were not quantified, in general, good communication improves patient satisfaction and adherence to therapy [79]. Such 3D models could therefore improve physician–patient communication.

Ultimately, the potential use of 3D-printed and bioprinted materials for retinal use is vast, from patient education to the seeding of RPE cells in a scaffold for the treatment of AMD. Specifically, with RPE cell implantations for wet AMD, long-term concerns such as the risk of infectious complications and the effectiveness of intervention versus anti-VEGF therapy still exist. For more clinical applicability, more in vivo animal and human trials are needed for most of the research discussed in this section.

## 4. Application of 3D Printing and 3D Bioprinting in Corneal Devices

Like the application of 3D printing and bioprinting in retinal applications, corneal applications must also be biocompatible, otherwise the risk of autoimmune responses, rejection, and infection exists. Given the versatility of 3D printing, possibilities in corneal applications are vast, including corneal tissue models for in vitro research, tissue grafts for keratoplasty procedures, contact lenses for improved drug delivery through the cornea, and corneal models for ophthalmologist training.

### 4.1. Corneal Modeling

The cornea is composed of five cell layers, as seen in Figure 3. The stroma, a collagen-rich layer, comprises the majority of the corneal thickness. Bowman’s layer and epithelium are found exterior to the stroma, and Descemet’s membrane and endothelium are on the interior stromal surface. Functionally, the cornea provides protection to the inner eye components and about two-thirds of the eye’s refractive power [80]. Some challenges with developing a cornea graft for research and transplant purposes include optical aberrations, sufficient tensile strength, and appropriate curvature.

The development of a synthetic corneal graft usable in humans is still in the early phases of research, but significant progress has been made in better understanding corneal cell behavior. For example, an increased slope gradient on a 3D-printed convex scaffold results in stronger adhesion and better alignment of rabbit corneal epithelial cell (RCEC) organization [81]. During the development of corneal scaffolds, curvature should be considered for epithelial strength, but balanced with the potential for optical alterations with excess curvature. A bioprinted cornea should have a similar tensile and compressive modulus to a human cornea. The long-term viability and optical properties of bioprinted corneas unable to replicate such qualities are highly uncertain [82]. Furthermore, the bioink used to 3D bioprint cells must also be biocompatible and non-toxic to allow future testing in humans. A novel, crosslinked, hyaluronic acid-based bioink was developed by Mörö et al., and through extrusion-based 3D printing a corneal graft was bioprinted. It showed effective ex vivo integration within a porcine cornea with no cytotoxicity detected [83]. Given its safety and effectiveness in bioprinting, this crosslinked hyaluronic acid bioink may have a role in bioprinting in vivo human corneal grafts.

### 4.2. Corneal Graft Development and Corneal Transplants 

#### 4.2.1. Keratoplasty

Keratoplasty, also known as a corneal transplant, is a surgical procedure to replace defective corneal tissue with a tissue graft. Multiple types of keratoplasty exist, each with different procedures and removal of diseased or damaged corneal layers [84]. Penetrating Keratoplasty is a full-thickness transplant, used when the full cornea is affected. Lamellar Keratoplasty (LK) is an overarching category for all partial thickness keratoplasty procedures. These include Deep Anterior Lamellar Keratoplasty (DALK), Descemet’s Stripping Endothelial Keratoplasty (DSEK), and Descemet’s Membrane Endothelial Keratoplasty (DMEK). DSEK and DMEK are subcategorized under LK as Endothelial Keratoplasty, which involves the replacement of posterior corneal layers. Lastly, keratoprosthesis is a full corneal removal followed by the implantation of an artificial cornea (Figure 4).

Corneal transplants require grafts from human donors [85], and this can pose significant challenges with the corneal donor material shortage [86], the limited graft survival time post-harvest, and the risk of graft rejection. The development of 3D-bioprinted corneas from collagen or other biocompatible materials may be able to address these concerns. Earlier research had shown potential for the 3D printing of a synthetic corneal graft. Isaacson et al. used a pneumatic 3D dual extrusion bioprinter to develop a structure resembling the corneal stroma. Using a collagen-based bioink, corneal keratocytes showed viability after 7 days. There was no consideration for long-term cell survival, cell–cell adhesion, and cell differentiation, and therefore it cannot be used as a model for disease research by itself. There was, however, significant promise and a precedent set for further testing and refining [87]. Since then, improvements have been made to bioink composition for better transparency, safety, and biocompatibility. Kim et al. developed a bovine cornea-derived extracellular matrix (Co-dECM)-based bioink with turbinate-derived mesenchymal stem cells (hTMSCs). The Co-dECM group caused the differentiation of hTMSCs into keratocytes which are needed for stromal generation. Transparency in the bioprinted cornea was calculated at 75% of the visible light spectrum passing through [88], whereas the human cornea has 80% to 94% light transmittance at 450 nm to 600 nm respectively [67]. The clinical significance of this difference in light transmittance remains unknown until in vivo human trials are conducted.

A current challenge with bioprinted corneal grafts is host integration and the collagen structure within the stroma. Using a graft developed with rabbit corneal epithelial cells (CEC) and adipose-derived mesenchymal stem cells (MSC), He et al. noted good integration within the cornea post-anterior lamellar keratoplasty [68]. The bioink formulated with GelMA (Gelatin methacrylate) and biocompatible PEGDA (long-chain poly(ethylene glycol) diacrylate) allowed for the appropriate growth and geometry of collagen fibrils. Without the orthogonal alignment of fibrils, the optical and mechanical properties of the cornea will be compromised. Alginate hydrogels are another common bioink material, though their rigidity can prevent cell proliferation and differentiation [69]. By altering the sodium citrate/sodium alginate ratios within the alginate gel, Wu et al. found that the matrix was more easily degraded by the extrusion-based 3D-printed human corneal epithelial cells. With easier degradation, cell proliferation capacity also improved [70]. These findings help improve alginate-based bioinks for corneal tissue development, but as no in vivo testing was done, the biocompatibility of alginate gels within the cornea is still uncertain.

#### 4.2.2. Keratoprosthesis

Keratoprosthesis is a surgical procedure involving full-thickness removal of the cornea and replacement with an artificial cornea. The most common type of keratoprosthesis is the Boston type I keratoprosthesis (BI-KPro), used when poor outcomes are predicted with penetrating keratoplasty, or with repeated corneal graft failures [71]. There are challenges with the existing materials used in BI-KPro—the rigidity of the polymethyl methacrylate (PMMA) and titanium can lead to elevated intraocular pressure, retroprosthetic membrane formation, corneal melt, and/or vitritis. BI-KPro also requires a corneal donor, which is difficult to obtain with the donor corneal tissue shortage [89]. Magalhães et al. developed a novel keratoprosthetic implant to bypass the need for a donor corneal graft and using the existing cornea. The implant materials were the same as for BI-KPro, but the titanium back plate was 3D-printed. In vivo implantation in a rabbit model led to significant post-surgical complications including retroprosthetic membrane development, elevated IOP, corneal melting, and infectious keratitis [89]. With these complications, its future application to humans seems unlikely, but a comparison to BI-KPro post-surgical issue incidence may give a better comparison of relative risk differences.

### 4.3. Corneal Regeneration

Boix-Lemonche et al. developed a mesenchymal stromal cell-loaded hydrogel using extrusion-based 3D bioprinting. The implant was placed into porcine eyes that had undergone Femtosecond-Laser-Assisted Intrastromal Keratoplasty. By day 14, although de novo extracellular matrix materials, including collagen, had been synthesized, the MSCs had migrated out of the implanted scaffold. As no healing occurred around the corneal flap at the site of the excisions [90], the quick MSC migration likely did not allow the stroma to fully regenerate. With stromal regeneration and collagen formation, the concern of collagen fibril directionality also exists. If a collagen-based bioink is used, shear stress, adjusted by the bioprinting nozzle diameter, can be used to help align the collagen fibril deposition [91].

### 4.4. Drug-Eluting Contact Lenses

High-resolution 3D printers can fabricate contact lenses (CL) or CL-like devices with built-in channels or reservoirs for drug delivery. Such devices are useful for conditions requiring prolonged drug release like dry eye disease, bacterial or allergic conjunctivitis, glaucoma, post-surgical antibiotic prophylaxis, and post-surgical anti-inflammatory treatment. A drug-eluting CL can provide benefits for patients with dexterity challenges while reducing medication loss through tears and nasolacrimal drainage permitted by sustained drug release. A timolol-loaded ethylene-vinyl acetate copolymer-polylactic acid (PLA/EVA blend) contact lens was 3D-printed via fusion deposition modeling by Mohamdeen et al. Unfortunately, the CL had poor drug release kinetics with 50% of the total drug released in the first 5 h. The reloadability of the CL drug reservoir was not mentioned, nor were any in vivo trials done to determine clinical significance [92]. Urbánek et al. developed a 3D-printed contact-lens-like drug delivery device (CLLD) embedded with levofloxacin and tetracaine and compared the drug release and penetration profile through the cornea versus topical eye drops ex vivo. The amount of released API (tetracaine/levofloxacin) from the CLLD was significantly greater compared to the predicted typical ocular absorption of the drugs from eye drops [93]. Although no in vivo testing of the device for infection control and prophylaxis was conducted, given the amount of API absorption measured, its clinical benefit would likely be very similar to commercially available eye drops.

### 4.5. Ophthalmologist Training

The 3D printing of ophthalmologist training materials can be a cost-effective and practical educational resource. A corneal trauma simulation model was developed using a multi-material polyjet 3D printer by Fu, Hollick, and Jones. In the event, 84.3% of participants agreed that the corneal trauma model was appropriately realistic for suture practice, and ophthalmologists self-reported an increase in confidence in corneal suturing after using the model [94]. This could be used as an alternative to animal or cadaver eyes where access and supply may be limited. For drug-eluting contact lenses and similar devices, 3D printing can allow fine-tuning of lens thickness, curvature, and microporosity for drug storage. These products may be close to in vivo testing and future commercialization given their safety and practicality. 

Significant challenges and concerns still remain with the in vivo use of bioprinted corneal tissue grafts, including graft cell viability, optical aberrations, graft rejection, poor re-epithelialization, and healing. Some of the studies discussed also had de-differentiation of corneal keratocytes back into mesenchymal stem cells which pose problems if stromal healing and regeneration have not been completed. Additional research and improvements are needed before a viable, bioprinted corneal tissue graft can be developed; however, significant progress has been made towards this goal in recent years (Table 6).

## 5. Applications in Glaucoma Therapeutics

Glaucoma, infamously referred to as the “silent thief of sight”, insidiously advances without noticeable early symptoms, often resulting in a delay in diagnosis and treatment. This disease, a leading cause of irreversible blindness worldwide, manifests as glaucomatous optic neuropathy. This pathology is identified by characteristic optic disc cupping or excavation, axonal degeneration, and apoptosis of retinal ganglion cells, leading to permanent vision loss [95]. The complex origins of glaucoma involve a combination of genetic and environmental factors. Elevated intraocular pressure (IOP) remains the only modifiable risk factor for the onset of glaucoma [95].

### 5.1. Diagnostic and Monitoring Tools

Three-dimensional printing technology has enabled the development of cutting-edge tools for researching, diagnosing, and monitoring glaucoma. A pioneering team developed a rapid, portable, and cost-effective glaucoma-detecting device consisting of a 3D-printed readout box and a smartphone camera, suitable for point-of-care settings, promoting the early identification of glaucoma [96]. Furthermore, 3D printing’s high resolution and customization capabilities with various ink materials can be applied to smart contact lens fabrication. One notable application involved the generation of 3D-printed smart soft contact lenses capable of constant IOP monitoring in a less invasive and convenient manner, substantially enhancing disease management, and raising the possibility of tailored personal treatment plans [97].

### 5.2. Minimal Invasive Glaucoma Surgery 

Current glaucoma treatment involves medication, generally in the form of eye drops and surgical procedures such as trabeculectomy, drainage implants, and minimal invasive glaucoma surgery (MIGS). The main objectives of these interventions are to reduce IOP and prevent further damage to the optic nerve. MIGS has been developed in recent years, involving the use of microsurgical devices, and making smaller incisions in the eye compared to traditional trabeculectomy. Three-dimensional printing technology has significantly advanced MIGS procedures requiring high accuracy and micro instrumentation [95]. For instance, Sverstad’s study tested a 3D-printed MIGS stent in vitro, featuring multiple chambers for modifiable aqueous humor drainage to better control IOP [98]. Another group employed 3D printing to develop a MIGS stent injector device, which is superior in offering precise placement and adaptability to various stent designs [99]. Furthermore, additive manufacturing of 3D printing layer by layer allows complex, structured, and multi-functional instruments to be printed into one piece without assembling. The customizable nature of 3D printing allows for the generation of instruments fitted to the patient and surgeon’s ergonomics. These tailored instruments may greatly improve outcomes and efficiency in minimized invasive surgeries by providing better control and precision, indirectly minimizing surgical errors [100,101,102].

### 5.3. Drug-Eluting Implants

The integration of 3D printing technology enables research teams to develop customizable and sustainable drug-eluting implants for glaucoma patients. Compared to traditional eye drop treatments, these implants offer convenience in long-term medication management, improve patient adherence, and potentially reduce side effects [92,103,104]. Also, 3D printing revolutionizes the production of drug-eluting contact lenses, offering high-resolution manufacturing for lenses with drug-storing apertures, ensuring sustainable drug release for an extended period. Current state-of-the-art 3D-printed drug-eluting contact lenses utilize printing methods such as hot melt extrusion and fusion deposition modelling to fabricate timolol maleate-loaded contact lenses [92,105]. An alternative drug delivery method for glaucoma patients, ensuring sustained drug release, involves punctal plugs with drug-eluting capabilities [106]. Researchers have been exploring high-resolution 3D printing techniques, such as digital light processing, to create drug-loaded punctal plugs with smooth surfaces [55]. The technology allows for localized delivery and precise customization of dosage administration for individual eyes, enhancing both comfort and personalized treatment [107]. In addition, a team demonstrated the possibility of using 3D-printed antimetabolite drug-release implants to prevent conjunctival fibrosis after glaucoma surgery [108]. However, most 3D-printed devices are still in the early stages of development and await further clinical trials. They face challenges such as a lack of standardization and regulatory guidelines and the limited availability of biocompatible materials. Nevertheless, more medical research teams are developing implants for glaucoma patients according to needs as 3D printing technology continues to advance and become more accessible [103,104,108].

In summary, the integration of 3D printing technology in glaucoma therapeutics offers various applications, including diagnostic tools, minimal invasive glaucoma surgery, and drug-eluting implants. Table 7 provides a summarized overview of the methods, parameters, results, and potential disadvantages/complications associated with each application.

## 6. 3D Printing in Ophthalmic Prototyping and Simulation 

### 6.1. Training Model for Surgical Practice

Three-dimensional printing technology offers realistic training devices that are more cost-effective and accessible than traditional animal or cadaver models. For instance, Lichtenstein’s team developed a 3D-printed tactile and structurally mimicking model incorporating both hard and soft tissues for orbital surgical training [109]. In a survey by Rama et al., more than 84% of students taught with 3D-printed orbital fracture models found them useful for enhancing visual–spatial skills, anatomy education, and surgical training [110]. Beyond orbital fracture training, 3D-printed eye models prove valuable for strabismus surgery training, with surveys confirming their fidelity compared to conventional rabbit head models. Silicone, identified as a better substitute material for extraocular muscles in cadaver eye models, along with the 3D-printed silicone head faceplate, enhances the realistic positioning and orientation of the eyeball compared to conventional rabbit head training [111]. As 3D printing models are based on tomography scans from various patients, they enable the exploration of diverse scenarios and anatomical variations. Additionally, 3D printing technology facilitates the fabrication of cost-effective teaching tools, including customized eye mounts for cadaveric eyes, artificial irises for wet lab training models, and an eyeball puzzle assembly toolkit, supporting personalized approaches to surgical education [112,113,114].

### 6.2. Prototyping and Simulation

3D printing is widely utilized in ophthalmological surgeries to enhance preoperative surgical planning, reduce surgical time, and improve surgical outcomes. A 1:1 patient-specific disease model can be fabricated using tomography scans, providing surgeons with a detailed representation of the individual’s distinct pathological condition. These rapid prototyping models allow surgeons to simulate and rehearse surgical steps, serving vital roles in surgical planning and indirectly reducing intraoperative decision-making time. Additionally, 3D-printed models can act as osteotomy or cutting guides and facilitate the pre-molding and pre-fitting of implants such as titanium meshes or plates, thereby reducing surgical time and indirectly minimizing surgical risks [115,116,117]. For instance, surgical corrections for craniofacial deformities, known for their high patient specificity and complexity, have benefited from 3D printing technology. Ouyang et al. reported a successful case of orbital hypertelorism correction with the aid of a 3D-printed cutting guide. The study created a 1:1 plaster model of the skull with a predetermined osteotomy guideline that assisted in surgical planning and served as an intraoperative guide [58]. Another team conducted forty cases of computer-assisted craniofacial malformity surgeries, treating hypertelorism, craniosynostosis, and orbital trauma with the aid of 3D-printed cutting guides. A stereolithographic model was fabricated for each patient, supporting complex cases of craniofacial osteotomy, surgical simulation, pre-planned optimized cutting guides, and pre-fitting of implants [118].

Beyond craniofacial malformities, 3D printing technology has been applied in treating tumor masses. For instance, performing radiosurgery for intraocular tumors, such as uveal melanoma, requires high radiation precision. Furdova’s team 3D-printed the patient’s eye model with the targeted tumor mass based on CT and MRI data to aid in planning stereotactic radiosurgery [119]. Three-dimensional printing also enables different structures of the eye, vasculature, and tumor mass to be printed in different colors and materials, with the model available in various scale sizes to better assist in visualizing and understanding the pathological region. Excitingly, the incorporation of 3D printing in veterinary ophthalmological surgeries, characterized by a more diverse anatomy and complexity, is now a possibility. Another study applied 3D printing in veterinary orbital and peri-orbital masses treatment planning and surgical planning, assisting in tumor resection, cryotherapy, and orbitotomy [120]. However, it is important to note the drawbacks of 3D printing, including the difficulty in simulating soft tissues and the potential for anatomical changes or tumor growth between the time of scanning and surgery, especially in young patients. There were also concerns about limited intraoperative flexibility once the implants were pre-fitted or pre-molded [118].

### 6.3. Use in Doctor–Patient Communication

In addition to its clinical and surgical applications, 3D-printed patient-specific models play a significant role in doctor–patient communication. Multiple studies suggest the benefits of 3D-printed models on the doctor–patient relationship, helping the patient understand their disease progress, surgery expectations and risks, and preventing potential malpractices [121,122,123].

## 7. Conclusions

In conclusion, the vanguard of ophthalmological innovation, championed by biomimetics, 3D printing, and bioprinting, heralds a bright and transformative future for the field. These technologies do not just imitate the complex architecture of ocular tissues but are unlocking unprecedented opportunities in oculoplastic surgery, retinal tissue engineering, corneal transplantation, and precise glaucoma interventions. Our comprehensive review underscores the significant strides made from 2014 to 2023, offering a panoramic view of the advances and setting a course for their clinical application.

With the development of state-of-the-art surgical models, enhanced preoperative planning, and innovative therapeutic devices, the promise of tailored ophthalmic care is closer than ever. As we look to the future, the potential advantages of 3D printing and bioprinting suggest a promising horizon for overcoming the limitations inherent in conventional methods, and the path forward is marked by a collaborative spirit that spans disciplines and bridges the gap between bench and bedside.

This synthesis of fundamental science and clinical practice invites a future where every patient benefits from the precision and personalization that these technologies offer. There is a palpable sense of optimism as we anticipate the continued evolution of these modalities, ensuring that the vision of individualized treatment becomes a reality. As we forge ahead, it is the unity of researchers, engineers, and clinicians that will illuminate the way, turning today’s innovations into tomorrow’s standard care in ophthalmology, enhancing lives one patient at a time.

## Figures and Tables

**Figure 1 biomimetics-09-00145-f001:**
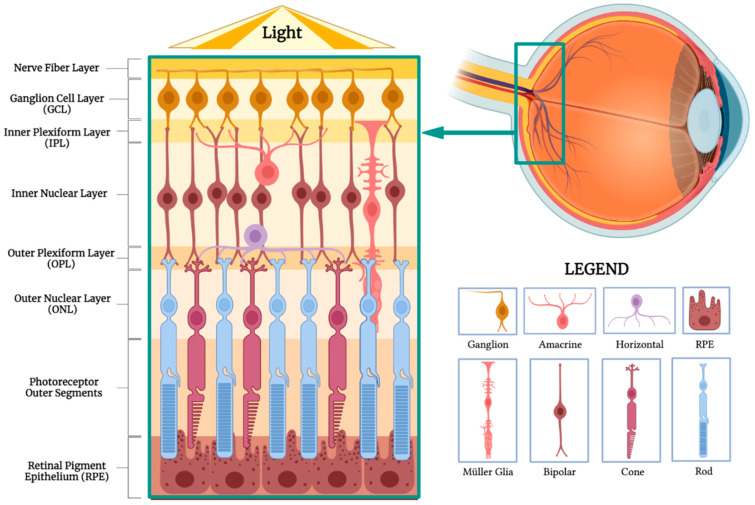
Structure and Composition of Retinal Layers (BioRender, https://app.biorender.com/, accessed on 26 January 2024).

**Figure 2 biomimetics-09-00145-f002:**
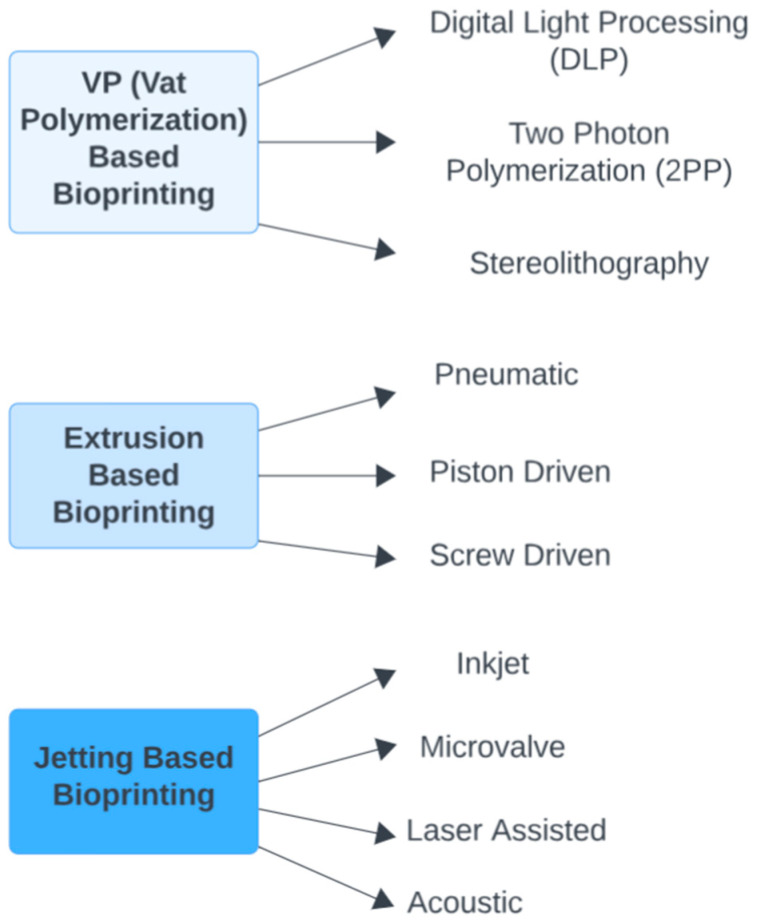
Bioprinting Technologies Used in Retinal and Corneal Applications.

**Figure 3 biomimetics-09-00145-f003:**
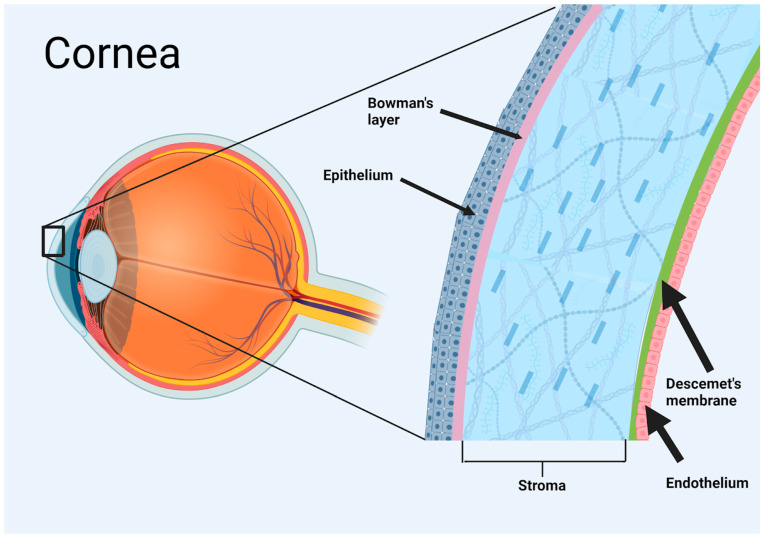
Structure of Corneal Layers.

**Figure 4 biomimetics-09-00145-f004:**
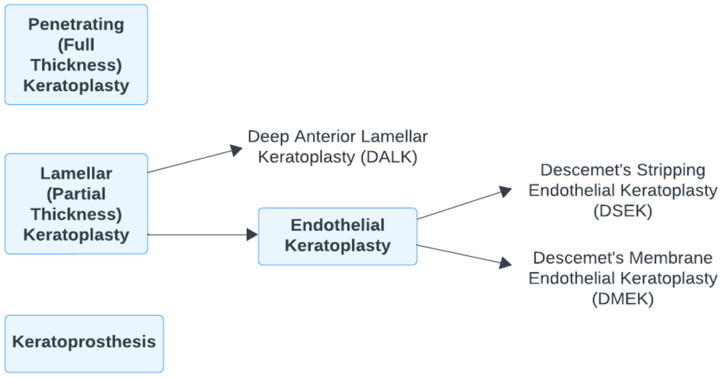
Categorization of various keratoplasty procedures.

**Table 1 biomimetics-09-00145-t001:** 3D Printing in Ocular Prothesis.

Usage	Description	Usage Studied	Key Features	Challenges	References
Direct	Directly 3D-printed	- Ocular prothesis - Orbital rehab post evisceration - Full-color ocular prothesis - Conformers - Scleral cover shell	- Fast production time - Personalization - Optimized weight and comfort - Cost varies based on material type - Stored data for fast edits and identical replacement - Semi-automatic, beneficial for less experienced ocularists - Potential to minimize patient visits for fittings	- CT radiation exposure; pregnancy and children - Long-term follow-up to determine the stability of these types of implants - Requires experience in 3D graphics and modeling software - Still a need for manual work in some usages, i.e., full-color prothesis - 3D models are rough, irritating the eyelid - Identify ideal biocompatible material	[8,10,11,12,16,20,21,22]
Guide	3D-printed model used to guide the production of a non-3D-printed prothesis	- Osseointegrated implant - Orbital prothesis - Migrated orbital implant	- Optimize surgery time - Optimal patient outcome - Precise, comfort, color matching, potential to improve global access	- Need for a detailed time and cost comparison to determine effectiveness of guides - 3D printing and casting are rate-limiting steps - Need for a larger study group and long-term follow-up	[15,17,18]

**Table 2 biomimetics-09-00145-t002:** Application of 3D Printing in Orbital Implants.

Usage	Description	Usage Studied	Key Features	Challenges	References
Template	Implant cut superimposed on the 3D model	- Pediatric orbital trapdoor fractures - Orbital floor	- Reduced surgery time - Increased accuracy and efficiency, with great fit - Low-cost option - Potential in endoscopic, minimally invasive surgery	- Need for technical and computational knowledge in fabricating the spline-based reconstruction mold	[32,33,34]
Mold	Implant molded onto the 3D model prior to surgery	- Zygomaticomaxillary fracture - Orbital blowout - Orbital wall and floor - Orbital cyst	- Individual and puzzle implants possible - Fast, easy, precise - No significant post-operative complications - Reduced operation time	- Sample size - Post-op eye protrusion significantly higher than pre-op - Uncertainties in virtual reconstruction of fractures	[35,36,37,38,39,40,43,44]
Pressing apparatus	Press real implant between the two sides of 3D models of the implant	- Orbital floor - Orbital wall	- Simple pressing action to obtain a custom implant - Cheaper alternative when compared to a fully 3D-printed implant	- Cost of the printer is a barrier to access - Possible difficulties in complex fractures - Small sample size	[41,42]
Direct printing	The implant itself is 3D-printed	- Inferomedial orbital strut - Orbital wall - Aesthetic - Orbital floor	- Simple and puzzle implants possible - Positive clinical outcomes and patient satisfaction - Stiffness of 3D-printed implant maintains structural stability	- Long planning, production, and sterilization time - Relies on access to a 3D printer - Lack of long-term follow-up data analysis	[45,46,47,48,49,50]

**Table 3 biomimetics-09-00145-t003:** 3D Printing for Nasolacrimal Stents.

Application	Usage	Key Features	Challenges	References
Lacrimal duct stent	Lacrimal duct bypass	- Material compatible with human tissue, no complications, mild inflammation that disappeared after 8 weeks	- Data from abstract, tested in vivo on rabbit models only	[55]
Magnetic micro-driller system	Duct recanalization	- 3D-printed parts for a microrobot using a magnetic actuation system - automated removal of blockage and built-in force sensor	- Automated path is unidirectional - Further research on ex vivo exploration before in vivo testing	[56]

**Table 4 biomimetics-09-00145-t004:** 3D Printing in Non-Surgical Ophthalmic Applications.

Application	Usage	Key Features	Challenges	References
Punctal plugs	Drug delivery	- Personalized for patient punctum size - Short production time, cost-effective	- Potentially less cytocompatible than controls - Need for clinical studies	[53,54]
Eye-lid crutches	Ptosis	- Low-cost - Easily removable and adjustable by the patient - Attaches to various frames - Flexible printing material allows for easier eye closure	- Non-universal design - Complications still present - Need for a large sample size and long-term research	[52]
Eye shield	Radiation	- 3D print model is a mold for silicone impression - Non-contact scanning, no use of anesthesia - Simplified laboratory work - Quick, easy, precise	- 3D scanner has difficulty capturing the precise site of the iris - Need for scanned data adjustment prior to print - Future printing of the shields themselves	[51]

**Table 5 biomimetics-09-00145-t005:** Summary of Case Report Data and Clinical Study Data.

**Procedure:**		**CAD Ocular Prosthesis**			**3D-Printed Orbital Prosthesis**			**3D-Printed Conformers**		**3D Printing in Orbital Floor Fracture Repair**							
Participants (n)		1	n.d.	3	3	10	1	9	5	14	1	12	22	1	4	28	1
Adverse Events	Severe	0	0	0			0			0	0	0	0		0	1	
	Non-severe	0	0	0			0								1		
	Pain			1													
	Infection			0		0						0		0			
	Pruritis			0													
	Dryness			2													
	Parasthesias									2							
	Enophthalmos											0		0	1	2	0
	Diplopia									1		0		1	0	1	
	Biochemical marker elevation					0											
	Inflammation					0											
	Functional/Cosmetic Problems							5	1		0			0	0		0
	Device Extrusion					0								0			
Observation Time (weeks)		24	4	4	n.d.	12	n.d.	156	16 (on average)	n.d.	n.d.	22 (median)	n.d.	4	Up to 24	26	n.d.
	References	[8]	[10]	[11]	[15]	[16]	[17]	[19]	[20]	[33]	[34]	[36]	[37]	[40]	[41]	[44]	[49]
**Procedure:**		**3D Printing in Orbital Wall Repair**							**Orbital Floor Reconstruction**		**3D Printing for Zygomaticomaxillary Fracture**		**3D-Printed Orbital Rim Reconstruction**		**3D Printing in Orbital Malformation**		
Participants (n)		82 (44 study grp.)			104	11	22	3	1		19		3		1		
Adverse Events	Severe	0			0	0	0	0	0								
	Non-severe				0	0			0								
	Pain										1						
	Infection					0	0				0		0				
	Degradation												1				
	Hypoesthesia	2			0						2						
	Enophthalmos				0	0		1									
	Diplopia	2			0		0	2			0		0		0		
	Inflammation						0										
	Functional/Cosmetic Problems												0		0		
	Device Extrusion					0											
Observation Time (weeks)		26 (minimum)			23.5		26	n.d.	26		26		26		n.d.		
	References	[38]			[39]	[42]	[46]	[48]	[43]		[35]		[47]		[50]		
**Procedure:**		**3D-Printed Conformers**							**Orbital Rim Reconstruction**		**3D-Printed Eye-Shield for Radiotherapy**		**3D-Printed Eyelid Crutches**		**Orbital Malformation Reconstruction**		
Participants (n)		9							3		2		1		4		
Adverse Events	Severe										0		0		0		
	Functional/Cosmetic Problems	5							1								
	Discharge	Common (but number not specified)															
	Scarring								1								
	Luxate/Extrusion	1															
Observation Time (weeks)		87 (mean)							Up to 1 year 9 months		22.5 days		22		43.5		
	References	[19]							[47]		[51]		[52]		[58]		

‘n.d.’ indicates a value not determined.

**Table 6 biomimetics-09-00145-t006:** Summary of 3D-Printed Corneal Devices.

Procedure	Material Used	Printing Type	Results	Setting	Advantage/Disadvantage	Reference
3D-printed implant scaffold for corneal regeneration	GelMA, type I collagen	Pneumatic, dual extruder printer	Increased slope gradient on the scaffold results in stronger adhesion and aligned cell organization.	In vitro	Understanding of how implant shape affects factors concerning corneal regeneration.	[81]
Corneal bioprinting utilizing collagen-based bioinks	Type I collagen-based bioink	Drop-on-demand (DoD) bioprinting	Cell viability confirmed at 7 days. Significantly less compressive modulus in printed vs. human cornea.	In vitro	Optical properties of printed cornea unknown.	[82]
Use of hyaluronic acid-based bioink for 3D printing of human corneal stroma	Human adipose stem cells (hASCs) and hASC-derived corneal stromal keratocytes, hyaluronic acid-based bioink with hydrazone crosslinking	Extrusion-based 3D printing	Development of a biocompatible bioink with future clinical potential and human testing.	In vitro & ex vivo	Bioprinted corneal structure showed effective ex vivo integration to porcine cornea. Potential for future in vivo human testing. No cytotoxicity detected.	[83]
3D bioprinting of a corneal stroma	Corneal keratocytes, methacrylated type I collagen, sodium alginate	Pneumatic 3D dual extrusion bioprinting	This is an earlier study trying to establish potential in the use of 3D printing for development of a corneal stroma.	In vitro	Keratocytes showed survivability and no toxicity noted. No consideration of long-term cell survival, cell adhesion, layering, and differentiation.	[87]
Development of a novel cornea-specific bioink	Bovine cornea-derived extracellular matrix) bioink	Not given	Bioink found to be biocompatible and allows for the differentiation of turbinate-derived mesenchymal stem cells (hTMSCs) and keratocytes.	In vitro	Biocompatible established in vitro, no cytotoxicity observed. Seemingly appropriate transparency of the printed cornea.	[88]
3D printing of an epithelium/stromal layer for an anterior lamellar keratoplasty	GelMA, long-chain poly(ethylene glycol) diacrylate, rabbit corneal epithelial cells, rabbit adipose-derived mesenchymal stem cells	Digital Light Processing (DLP)	Bioprinted scaffold (epithelial and stromal layer) integrates into the existing rabbit cornea well leading to re-epithelialization and stromal regeneration.	In vivo	Biocompatible, good in vivo integration and potential for future human testing.	[68]
Finding an appropriate sodium citrate/sodium alginate ratio for bioprinting corneal cells	Alginate-based bioink and human corneal epithelial cells	Extrusion-based 3D cell-printing	By altering the sodium citrate/sodium alginate ratio in the gel, its ability to degrade improves, allowing better corneal cell proliferation.	In vitro	High cell viability was maintained.	[70]
Novel method for Keratoprosthesis using 3D printing and the recipient’s own cornea	3D-printed titanium back plate, PMMA front stem	Not given	Successful technique, improving on the standard keratoprosthesis procedure which requires donor corneal tissue.	In vivo	Many post-surgical complications.	[89]
Use of 3D-bioprinted scaffolds with mesenchymal stromal cells for a keratoplasty procedure	3 different multipotent mesenchymal stromal cells (adipose-derived, bone marrow-derived, and corneal stroma-derived)	Extrusion-based 3D bioprinting	Femtosecond-Laser-Assisted Intrastromal Keratoplasty was highly effective for corneal excision. The keratoplasty procedure and bioink did not undergo appropriate healing and cell differentiation.	In vivo	Mesenchymal stromal cells did not undergo differentiation towards corneal keratocytes. Poor healing around implanted corneal flap.	[90]
Development of a 3D-printed drug-eluting contact lens	Ethylene-vinyl acetate and copolymer-polylactic acid blend	Fusion deposition modeling	Successful development of a drug eluting contact lens, but poor pharmacologic characteristics.	In vitro	Poor drug release kinetics (majority of drug released in the first 24 h—poorly sustained, long-term release kinetics).	[92]
Development of a 3D-printed drug-eluting contact lens	Collagen based material, levofloxacin, tetracaine	Not given	Successful production of a drug-eluting contact lens embedded with levofloxacin and tetracaine.	In vitro	Released API from the drug delivery device was significantly greater vs. the predicted typical ocular absorption of the drug from eye drops. However, clinical significance was not measured.	[93]

**Table 7 biomimetics-09-00145-t007:** Application of 3D Printing in Glaucoma Therapeutics.

Usage	3D Printing Methods Used	Results	Types of Study	Challenges	Reference
Diagnosis & Monitoring	3D-printed (printer not specified) readout box and lateral flow assay (LFA) case	Rapid, portable, and cost-effective glaucoma-detecting device by LFA quantification of BDNF concentration in artificial tear fluids; suitable for point-of-care settings	Prototype development and characterization	Limited experimental detection limit; limited stability information	[95]
	Automated nozzle injection system (Nordson EFD) equipped on a three-axis computer-controlled translation stage	3D-printed smart soft contact lenses for constant IOP monitoring; intrinsic properties unchanged compared to commercial soft contact lens: biocompatibility, softness, transparency, wettability and oxygen transmissibility	in vivo	-	[96]
MIGS	Micro-precision three-dimensional printer with projection micro stereolithography technology and photosensitive materials	Proof of concept for a glaucoma stent with multiple lumina that can be separately opened with an argon laser trabeculoplasty (ALT)-like procedure for a predictable pressure-lowering effect	in vitro	Further in vivo study is suggested	[97]
	A microstent injector device was manufactured using a fused deposition modelling (FDM) 3D printer (Form 2 printer, Formlabs Inc) and various photoactive polymers	Development of a 3D-printed antifibrotic drug-eluting MIGS stent with modifiable aqueous humor drainage.	in vitro	Limited in vivo, invitro and clinical data of the device; further optimization of the drug-eluting coating is required	[98]
	3D-printed multi-steerable cable-driven instrument printed using Digital Light Processing (DLP) (Perfactory1 Mini XL, EnvisionTEC GmbH)	Fully 3D-printed, customizable instruments for better control and precision based on ergonomic principles. Potential applications in MIGS	Comparison studies: questionnaires and task performances; instrument mechanical evaluation	Limited availability of biocompatible materials; familiarization of instrument required to avoid excessive force and instrument breakage	[99,100,101]
Drug-Eluting Implants	Triamcinolone acetonide-loaded polycaprolactone-based ocular implants fabrication using GeSiM 2.1 Bioscaffolder 3D Bioprinter	Customizable (in shapes and drug loadings); sustainable drug-eluting implants (6 months); biocompatible, safe ocular application (>90% cell viability)	in vitro	Early stage of development; further in vivo studies and clinical trials are required to ensure the safety of implants	[102]
	The device is not 3D-printed; the paper brought up the possibility of PCL-based drug delivery implant	Developed an intracameral polycaprolactone glaucoma device (spin-casting made PCL thin films encapsulating proprietary hypotensive agent); long-term (23 weeks), effective IOP reduction with drug-eluting implants; 3D-printed injector device for precise placement.	in vivo (rabbit eye model)	-	[103]
	3D-printed PCL and chitosan-based drug-eluting implant using heat extrusion technology	Sustained, long-term (8 weeks) 5-fluorouracil drug-releasing implant; effective in suppressing fibroblast contractility and preventing conjunctival fibrosis after glaucoma surgery; biocompatible (no significant changes in cell viability) and biodegradable	in vitro	In vivo experiments required; risks of allograft rejection for biomaterials	[107]
	Hot melt extrusion coupled with fusion deposition modelling (FDM) to print ethylene-vinyl acetate copolymer–polylactic acid blends	Sustained drug release of timolol maleate for extended periods (3 days); introduction of drug-eluting contact lenses with high-resolution 3D printing manufacturing	Prototype development and characterization	Drug release optimization, in vitro and in vivo studies needed	[91]
	Modified commercial inkjet printer (O2Nails V11 inkjet printer) and Form 1+ 3D printer with v4 clear resin (Formlabs Inc., MA, USA)	Contact lenses drug-release over at least 3 h, longer than eye drops; accurate drug dose-loading quantification with near-infrared (NIR) spectroscopy	in vitro	-	[104]
	Digital light processing (DLP) 3D printing was employed to assemble polyethylene glycol diacrylate (PEGDA) and polyethylene glycol 400	Punctal plugs with drug-eluting capabilities for sustained drug release (dexamethasone, 7 days)	in vitro	Drug-loaded micro stents were less cytocompatible than blank controls	[106]
	3D-printed polypills by selective laser sinter (SLS) printer	Developed a non-destructive method for quality control for 3D-printed antimetabolite drug-release implant (the drugs loaded were amlodipine and lisinopril for preventing conjunctival fibrosis)	Prototype development and characterization	Lack of standardization and regulatory guidelines for drug-eluting implants	[55]

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
