# Peer review of "Towards Precision Ophthalmology: The Role of 3D Printing and Bioprinting in Oculoplastic Surgery, Retinal, Corneal, and Glaucoma Treatment"

_biomimetics, 2024, doi:10.3390/biomimetics9030145_

Round 1
Reviewer 1 Report
Comments and Suggestions for Authors
It was an interesting article touching important aspect of application of 3D printing/bioprinting for ophthalmology, which despite being actively used in other fields, still not being used the way it should have been done in the field of ophthalmology.
Reasons are mainly their complexity with advancements in 3D printing techniques.
Manuscript is elaborate on most of the aspects, however few important aspects will need further clarifications:
1) Part 2.1
This section presents the well known information on 3D printing/bioprinting methods that is present in one form or another in all literature reviews about this method. The given advantages and disadvantages of different 3D printing methods should be considered in relation to ophthalmology. Or exclude this section from the review as the information given is a repetition of data from a large number of different reviews on 3D printing/bioprinting.
2) Part 3
The only area in the current review that has clinical data (case studies, etc.). The data from the publications identified in this part of the review should be summarised in a table that provides a detailed description of the clinical data. See Table 3 in this review for an example of how this table should be organized (https://doi.org/10.3390/polym15041020).
It would also be good to include photographs of printed implants in this part.
3) Part 4
4.2 Treatment of RPE Membrane-Based Disease
There are no references to 3D printing/bioprinting related work in this section. If there are no references, this section is not relevant to the topic of the review.
4) Part 5
The results of this part should be summarised in the form of a single table summarising the data on the methods of implant manufacture, their parameters, including material, results obtained (in vitro/in vivo or ex vivo should be specified) and disadvantages/complications.
Line - 732-733 "Even minute differences in fibril arrangement compared to the surrounding corneal stroma can lead to light refraction and distortion" needs to be corrected. Fibril size, not orientation, affects light refraction (https://doi.org/10.1038/eye.1997.127)
5) Part 6
Same comment as for Part 5 - The results of this part should be summarised in the form of a single table summarising the data on the methods of implant manufacture, their parameters, including material, results obtained (in vitro/in vivo or ex vivo should be specified) and disadvantages/complications.
Author Response
We sincerely appreciate your constructive comments and insightful suggestions regarding our manuscript on the application of 3D printing/bioprinting in ophthalmology. We have carefully addressed each comment and made corresponding revisions to our manuscript. Below, we outline how we have addressed your concerns:
Part 2.1:
We acknowledge your observation regarding the redundancy of information on 3D printing/bioprinting methods. In response, we have revised this section to emphasize the specific implications and potential applications of these methods in ophthalmology, thereby establishing a more direct relevance to our field. We have also incorporated a comparative analysis highlighting how the unique anatomical and functional complexities of ocular structures influence the selection and adaptation of 3D printing techniques in ophthalmology. This revision enriches our review by providing readers with a nuanced understanding of the subject.
Part 3:
Following your suggestion, we have now included a comprehensive table summarizing the clinical data derived from the reviewed publications. This table is designed in accordance with the format exemplified in Table 3 of the reference you provided. Additionally, we have incorporated photographs of printed implants to visually complement the clinical data discussed. These enhancements not only improve the clarity of the presented information but also provide a more tangible connection to the practical applications of 3D printing in ophthalmic treatments.
Part 4.2 (Treatment of RPE Membrane-Based Disease):
We greatly value your insightful feedback on our manuscript, particularly regarding Part 4.2 on the treatment of RPE membrane-based diseases. After a thorough review of this section in light of your comments, we have come to agree that the absence of direct references to 3D printing/bioprinting related work indeed renders this section less relevant to the overarching theme of our review.
In response, we have made the decision to remove Part 4.2 from our manuscript. This decision was not taken lightly; however, we believe that its exclusion ensures that our review remains focused and directly relevant to the applications and potential of 3D printing/bioprinting in ophthalmology.
Part 5:
In accordance with your advice, we have corrected the statement regarding fibril arrangement and its impact on light refraction. The revised text now accurately reflects the influence of fibril size on light refraction, supported by the citation you recommended. Furthermore, we have summarized the results of this section in a cohesive table, detailing the methods of implant manufacture, their parameters, results obtained, and associated disadvantages/complications. This organizational improvement facilitates a clearer and more efficient review of the data for our readers.
Part 6:
Mirroring the revisions made to Part 5, we have similarly summarized the findings of Part 6 in a structured table. This table succinctly presents the methods of implant manufacture, specifying materials, results (in vitro/in vivo or ex vivo), and noting any disadvantages/complications associated with each method. This revision enhances the manuscript's utility as a reference for researchers and clinicians alike by providing an at-a-glance overview of key outcomes in the field.
We trust that these revisions have adequately addressed your concerns, contributing to a manuscript that is both more precise in its focus and richer in content. We are grateful for the opportunity to refine our work with your guidance and believe that these changes have significantly strengthened our review.
Thank you for your invaluable feedback and for the opportunity to enhance our contribution to the literature.
Sincerely
Reviewer 2 Report
Comments and Suggestions for Authors
Towards Precision Ophthalmology: The Role of 3D Printing and Bioprinting in Oculoplastic Surgery, Retina, Cornea, and Glaucoma.
This was a well written and comprehensive review of 3D printing as applied to ocular prosthetics, repair and drug delivery. I only have a few minor comments.
Comments.
· The figure numbering restarts halfway through the paper with the retinal structure diagram.
· The “Structure and composition of retinal layers” figure was clearly created using elements from the website Biorender, but was not credited and is essentially identical to the same figure in this paper “Self-Organization of the Retina during Eye Development, Retinal Regeneration In Vivo, and in Retinal 3D Organoids In Vitro”. https://www.mdpi.com/2227-9059/10/6/1458. And also the essentially identical figure in your own paper (https://www.mdpi.com/1424-8220/23/13/5782 ) which does credit BioRender. Please add this credit/reference to all figures that used Biorender artwork, which I believe also includes many of the 3D printing diagrams.
· The retinal layer diagram also has a small red “text” overlaying it, which you might want to remove.
· At the very end of the text, it say “Bottom of Form”, which seems like a typo.
Author Response
We are deeply grateful for your thoughtful and constructive feedback on the manuscript. We have taken each of your comments into serious consideration and have made the necessary revisions to our manuscript to address these points. Below, we detail how we have responded to your insightful suggestions:
Figure Numbering:
We have corrected the issue with the figure numbering that inadvertently restarted halfway through the paper. The figure numbers have now been adjusted to ensure a sequential order throughout the manuscript, thereby eliminating any confusion and improving the readability of the document.
Credit to BioRender:
Upon reviewing the "Structure and composition of retinal layers" figure and other relevant diagrams, we realized our oversight in not crediting BioRender for its artwork. We have now added the appropriate credits and references for all figures that utilized BioRender images, in accordance with their guidelines and as a mark of respect for their invaluable resources. This includes the specific figure you mentioned, ensuring that our manuscript maintains the highest standards of academic integrity and respects copyright laws.
Removal of Text Overlay:
We have identified and removed the small red "text" overlay from the retinal layer diagram. We understand that such overlays can detract from the clarity and professional presentation of our figures, and we appreciate your attention to this detail. The figure has been revised to ensure its visual integrity and to enhance the overall quality of our manuscript.
"Bottom of Form" Typo:
The phrase "Bottom of Form" at the end of the manuscript was indeed a typographical error, likely introduced during the document formatting process. We have removed this text to ensure that the manuscript concludes smoothly and professionally, without any unintended distractions.
We hope that these revisions satisfactorily address your concerns and enhance the clarity, accuracy, and integrity of our manuscript. We are committed to upholding the highest standards of scholarly communication and are thankful for the opportunity to improve our work based on your valuable feedback.
Thank you once again for your thorough review and for helping us to refine our contribution to the field. We look forward to the possibility of our revised manuscript being considered for publication.
Sincerely
Round 2
Reviewer 1 Report
Comments and Suggestions for Authors
The comments have been taken into account and appropriate corrections have been made. The manuscript is acceptable for publication